# Does Potassium Modify the Response of Zinnia (*Zinnia elegans* Jacq.) to Long-Term Salinity?

**DOI:** 10.3390/plants12071439

**Published:** 2023-03-24

**Authors:** Hanna Bandurska, Włodzimierz Breś, Małgorzata Zielezińska, Elżbieta Mieloszyk

**Affiliations:** Department of Plant Physiology, Poznań University of Life Sciences, Wołyńska 35, 60-637 Poznań, Poland

**Keywords:** salinity, decorative value, membrane injury, proline, lignins

## Abstract

Salinity is one of the major abiotic stress factors hindering crop production, including ornamental flowering plants. The present study examined the response to salt stress of *Zinnia elegans* ‘Lilliput’ supplemented with basic (150 mg·dm^−3^) and enhanced (300 mg·dm^−3^) potassium doses. Stress was imposed by adding 0.96 and 1.98 g of NaCl per dm^−3^ of the substrate. The substrate’s electrical conductivity was 1.1 and 2.3 dS·m^−1^ for lower potassium levels and 1.2 and 2.4 dS·m^−1^ for higher potassium levels. Salt stress caused a significant and dose-dependent reduction in leaf RWC, increased foliar Na and Cl concentrations, and reduced K. About 15% and 25% of cell membrane injury at lower and higher NaCl doses, respectively, were accompanied by only slight chlorophyll reduction. Salt stress-induced proline increase was accompanied by increased P5CS activity and decreased PDH activity. More than a 25% reduction in most growth parameters at EC 1.1–1.2 dS·m^−1^ but only a slight decrease in chlorophyll and a 25% reduction in the decorative value (number of flowers produced, flower diameter) only at EC 2.3–2.4 dS·m^−1^ were found. Salt stress-induced leaf area reduction was accompanied by increased cell wall lignification. An enhanced potassium dose caused a reduction in leaf Na and Cl concentrations and a slight increase in K. It was also effective in membrane injury reduction and proline accumulation. Increasing the dose of potassium did not improve growth and flowering parameters but affected the lignification of the leaf cell walls, which may have resulted in growth retardation. *Zinnia elegans* ‘Lilliput’ may be considered sensitive to long-term salt stress.

## 1. Introduction

Soil salinity is one of the major stress factors affecting plants growing in natural environments and those cultivated by humans [1]. It is widespread in arid, semi-arid, and coastal regions with poor soil water resources caused by low rainfall and high evapotranspiration [2]. However, it also arises from the natural weathering of saline rocks, anthropogenic activities such as inappropriate irrigation practices with salt water, and persistently changing climatic conditions [3,4,5]. Nowadays, almost 10% of total land and 50% of irrigated land are affected by salinity [6]. Soils are considered saline when the electrical conductivity (EC) of the saturated soil extract is 4 dS m^−1^ or higher. It is an equivalent of 40 mM NaCl and gives an osmotic potential of approximately −0.2 MPa [4,7]. Soil salinity is also an increasing threat to the ornamental plants used in gardening and landscaping and as cut flowers [8]. This is a significant handicap for urban green belts because soil salinity in these areas is common due to the high salt content caused by the use of NaCl in the deicing of roads and sidewalks [9,10,11]. Salt-affected soil is becoming a serious problem in landscaping and gardening because of the diminishing sources of high-quality water, which requires the re-use of wastewater for the irrigation of green areas in the urban environment as well as in gardens and fields [8,12]. Salt stress is also an issue in the production and cultivation of potted ornamental plants. Large amounts of salt may often be present in potting substrates made from a mixture of different materials [8]. Moreover, bedding plants produced in containers with regular watering usually have large leaf areas and are root-bound. When they grow in saline soil, they quickly use the water from the root ball but are frequently unable to obtain water from the surrounding soil and suffer from dehydration [13]. Ornamental plants used in landscaping and gardening significantly differ in their susceptibility to salt stress [12,14]. 

Zinnia (*Zinnia elegans* Jacq.) is one of the most popular annual ornamental plants. It has relatively low requirements, can adjust to most soil conditions (clay, chalk, sand, loam), and tolerates alkaline, neutral, and even acid soils [15,16]. This species is cultivated worldwide as bedding plants in gardening and landscaping and as cut flowers. Moreover, the low requirement and decorative value of zinnia (lively and uniform flower colors) make it applicable in vertical gardens coverage and garden roofs. The wide use of zinnia and more common salt stress in the cultivation of ornamental plants makes salt resistance an important feature. The assessment of salt resistance in ornamental plants should include both quantitative data (biomass production, growth parameters) and qualitative analyses, including flower production and overall plant appearance [17]. The existing literature data show a great cultivar variability in resistance to salinity between zinnia species and cultivars. Villarino and Mattos [14] revealed that *Zinnia angustifolia* ‘Star Gold’ exhibited significant growth inhibition at an EC of 6 dS m^−1^ and 100% mortality at an EC of 14.2 dS m^−1^. Similarly, the cultivars of *Zinnia marylandica* and *Zinnia maritima* were shown to be sensitive to salt stress based on growth reduction and mortality. The shoot dry weight in all examined cultivars was reduced at an EC of 4.2 dS m^−1^ by 50% to 56%, and plants died at an EC of 6.0 to 8.2 dS m^−1^ [18]. Some *Zinnia elegans* cultivars were slightly more resistant to salinity. Zinnia ‘Dreamland’ irrigated with varying concentrations of NaCl for 5 weeks showed reduced growth without affecting flowering. Plants can survive NaCl concentrations that produce EC of the potting medium at 12 dS m^−1^ but show severe leaf injury and water stress symptoms [19]. However, *Zinnia elegans* ‘Magellan’ was resistant to moderate salinity (EC below 10 dS m^−1^), had flowers reduced by 75% at an EC of 10 dS m^−1^, and died at an EC of 15 dS m^−1^ [20]. Escalona et al. [21] showed that salt stress at an EC level of 6 dS m^−1^ caused a decrease in biomass in *Zinnia elegans* but did not affect flower production and toxicity symptoms in leaves. The decline in growth parameters was shown in *Zinnia elegans* ‘Benary’s Giant Salmon Rose’ and ‘Benary’s Giant Golden Yellow’ irrigated with saline water at an EC of 10 dS m^−1^. However, marketable cut flowers based on stem length were produced [22]. 

The detrimental effect of soil salinity is water deficit in plants caused by the reduction in water availability (physiological drought) as well as the toxic effect of ions, especially sodium (Na^+^) and chloride (Cl^−^), caused by their excessive uptake [23]. Thus, plant responses to salt stress were divided into two phases. The first phase is caused by the restriction of water availability, takes place within minutes to days, and includes stomatal closure and the inhibition of cell expansion, mainly in the shoot, leading to growth reduction. The second phase takes place over days or even weeks and is caused by excessive toxic ions in plant cells, which harm metabolic processes [24]. The response to salinity is complex and involves numerous adjustments at morphological (early flowering, growth inhibition, prevention of lateral shoot development, root adaptations), physiological (stomatal closure, osmotic adjustment, Na/K discrimination), and biochemical levels (antioxidant activity, change in hormone level, increased proline level), which help plants cope with stress [23,24,25]. 

Growth inhibition and a reduction in the total leaf area under salinity conditions are the effects of lower cell turgor and cell wall extensibility caused by increased lignin deposition [26]. The inhibition of leaf growth in drought- and salt-stressed plants can be considered a component of a stress coping strategy. It is an avoidance mechanism that reduces water loss via transpiration under saline conditions and may prolong plant survival [27]. In the production of ornamental plants, it could be beneficial for nursery growers, especially when they want to produce compact plants and avoid using plant growth regulators [28].

Proline accumulation is another example of the salt stress coping strategy [25,29]. This amino acid may be involved in stress avoidance and stress tolerance strategies. Proline accumulation under salinity stress conditions decreases cells’ osmotic potential and allows for water absorption. Besides acting as an osmolyte, proline is a membrane and protein stabilizer, as well as a free radical scavenger. It protects plant cells against the detrimental effects of dehydration and the accumulation of toxic ions [25,30]. The proline level in plant cells is controlled by several cellular mechanisms responsible for its synthesis and degradation [25,29]. This amino acid can be biosynthesized through glutamate and ornithine pathways in the cytosol. In salt-stressed plants, proline is primarily synthesized from glutamate that is converted to glutamic-γ-semialdehyde (GSA) by Δ^1^-pyrroline-5-carboxylate synthetase (P5CS). GSA is spontaneously cyclized to Δ^1^-pyrroline-5-carboxylate (P5C), which is reduced to proline by pyrroline-5-carboxylate reductase (P5CR). The proline degradation pathway occurs in mitochondria and is catalyzed by two enzymes. Proline dehydrogenase (ProDH) catalyzes proline oxidation to P5C, which is converted into glutamic acid by P5C dehydrogenase (P5CDH).

Salinity affected by a high concentration of NaCl in the soil solution resulted in an excessive accumulation of Na and Cl and reduced the uptake of other elements, including potassium, which plays an essential role in many physiological and biochemical processes [31]. One of the methods of alleviating the negative effect of salt stress caused by excess sodium chloride is fertilization with an increased dose of potassium [32]. A hypothesis that potassium application can reduce the harmful effects of salinity on plant growth and development was proposed by Ben-Hayyim et al. [33], and it was confirmed by, among others, Tzortzakis [34], Umar et al. [35], Amjad et al. [36], and Hashi et al. [37]. However, no effect of K fertilization under saline conditions or even a negative effect of K nutrition on salt resistance were reported [38,39].

The present study investigates whether the application of an increased potassium dose could ameliorate the harmful effect of salinity on *Zinnia elegans* ‘Lilliput.’ The assessment was made based on measurements of growth parameters, morphological features, and physiological and biochemical parameters. The choice of this variety was not accidental. Plants of this cultivar grow to a height of 20–25 cm, so they are very suitable for growing on flower beds in urban areas, especially along streets and sidewalks. The application rates of sodium were determined based on earlier studies and our preliminary experiment. For example, in Poland’s urban areas, soil salinity rarely exceeds an EC of 3 dS m^−1^ [40,41,42,43], while in Canada, 50% of the electrical conductivity of soil along roads exceeded 2 dS m^−1^ [44]. According to Kotuby-Amacher [45], the growth of sensitive ornamental plants and grasses may be restricted when the soil EC exceeds 2 dS m^−1^ (e.g., China aster, geranium, annual bluegrass, Kentucky bluegrass). 

## 2. Results

### 2.1. Results of the Preliminary Experiment

The results of the preliminary experiment showed that zinnia ‘Lilliput’ is sensitive to salinity. One-way ANOVA results showed a significant effect of salt stress on the examined growth parameters (Table 1). A gradual and substantial reduction in all examined growth parameters was observed as the NaCl dose increased (Figure 1). A negative effect of sodium chloride on the plant height, leaf area, fresh weight of the aboveground part, and diameter of the inflorescence on the main shoot (i.e., the features determining the ornamental value) was found just at the substrate EC of 1.67 dS m^−1^ (1.47 g NaCl).

Moreover, a significant percentage of plants grown in the substrate with higher doses of sodium chloride (1.98, 2.48, and 2.99 g dm^−3^) showed typical visual symptoms of saline injury such as scorching and necrosis around the leaf margins, extending to the whole blade over time and ultimately leading to leaf drop. The first symptoms of injury were observed after 22, 19, and 18 days of cultivation in substrates supplemented with 1.98, 2.48, and 2.99 g NaCl, respectively. The defoliation and final death of some plants were noted after 26, 25, and 24 days. Finally, 30 to 40% of the plants died. In the control treatment and in combinations with the three lowest doses of NaCl, no damage symptoms on the plant were observed. The presented results were the basis for selecting the range of salt concentrations used in the next experiments.

### 2.2. Results of Main Experiments

#### 2.2.1. Water Management Parameters

A statistically significant effect of substrate salinity on water loss from containers was shown, as well as the effect of the date (days of the experiment) and potassium dose and their interaction (Table 2).

The water loss from containers was significantly higher in the control combination than in both stress combinations (Figure 2). Significantly less water evapotranspired from the containers with plants exposed to a higher NaCl dose. This shows that plants grown in the saline substrate can uptake less water than those produced without salt.

Increasing the dose of potassium resulted in a slight reduction in water evaporation, especially at the beginning of the experiment in plants grown without salt and with a lower NaCl dose (Figure 3).

The restriction of water availability in salt-stressed plants affected the water balance, as evidenced by the statistically significant NaCl dose- and time-dependent changes in RWC. An interaction between treatments in terms of RWC was not found (Table 1). The greater the salt dose, the greater the leaf RWC reduction (Figure 4). Statistically significant differences in RWC were also noted between the dates of measurements. In both the control and salt-treated plants, the RWC was higher on the second and third sampling dates compared to the first one. Plants supplemented with higher potassium doses showed a statistically significant higher decrease in RWC, except those grown with the highest NaCl dose on the third sampling date. On this date, the leaf RWC in plants supplemented with a higher potassium level was higher than that in those supplemented with a lower dose. 

#### 2.2.2. Na, Cl, and K Concentrations in Leaves

The two-way ANOVA showed that adding NaCl to the substrate caused statistically significant changes in leaf Na, Cl, and K concentrations (Table 3). In the leaves of control plants, the Na was at about 0.13% and was much lower than the Cl concentration (about 1.5%). The applied salt stress caused a significant and dose-dependent increase in Na and Cl concentrations (Figure 5A,B). In leaves of plants growing with the higher NaCl dose, the Na concentration increased threefold, and Cl increased about twofold. The concentration of K in control plants was about 5.75%. Salt stress caused a statistically significant and dose-dependent decrease in K in leaves (Figure 5C). About a 5% and 11% decrease was shown with the lower and higher doses of NaCl, respectively. As a result of these changes, salt stress caused a reduction in the K/Na ratio in leaves from 48.16 to 17.38.

The addition of a higher potassium dose to the substrate caused a significant but slight increase in this element in the plant leaves from all combinations (Figure 5C) and an increase in the K/Na ratio. The concentration of Cl in the leaves of plants grown with the higher potassium dose was significantly but slightly lower than that in those grown with a lower dose (Figure 5B). However, the higher potassium dose was effective in the reduction in Na accumulation in leaves. Na was reduced by about 22% and 37% in plants exposed to lower and higher NaCl doses, respectively (Figure 5A). The interactive effect of NaCl and potassium on the concentration of Na in leaves was also observed (Table 3).

#### 2.2.3. Growth Parameters

Analyses of variance indicated a significant effect of salinity on the estimated growth parameters (Table 4). A significant effect of the potassium dose on the changes in the main shoot height, lateral shoot number, leaf area, and inflorescence diameter was found. However, the interaction of NaCl and potassium treatment was found only in the main shoot height and leaf area changes.

Salt stress affected the reduction in all growth parameters (Figure 6). The reduction was greater with the increasing salt dose applied (Figure 6A–F). Higher potassium doses significantly reduced the main shoot height, the number of lateral shoots, the leaf blade area, and the inflorescence diameter (Figure 6A–E). In addition, the potassium-induced reduction in the mean shoot height and leaf area was higher in plants supplemented with a higher NaCl dose (Figure 6A,C).

#### 2.2.4. Chlorophyll, Lignin, and Membrane Injury Index

The two-way ANOVA indicated a significant effect of salt stress on the content of leaf chlorophyll and lignin as well as the membrane injury index (Table 5). Potassium significantly affected the lignin content and membrane injury index but not the chlorophyll content. The interaction of these factors on the content of chlorophyll and lignin and membrane injury was not revealed.

An increase in salinity led to a decrease in the leaf chlorophyll content (Figure 7A). The lignin content in the cell wall increased significantly with the increase in NaCl. Higher lignin levels were found in the leaves of plants supplemented with higher potassium doses (Figure 7B). In addition, membrane injury increased with increasing salinity. However, it was lower in plants supplemented with higher potassium doses (Figure 7C). 

#### 2.2.5. Proline Content and Enzyme Activity

The three-way ANOVA showed a significant effect of the salinity, the sampling date, and their interaction on the proline content and the activity of both enzymes (Table 6). Moreover, a significant effect of potassium and the interaction between NaCl × potassium and NaCl × date × potassium on proline content and PDH activity was revealed.

Salt stress affected the time, dose, and potassium-dependent increase in leaf proline content (Figure 8A). On the first date, the proline content increased with the salinity increase and was the highest in plants grown with higher NaCl (1.98 g dm^−3^) and potassium doses (300 g dm^−3^). A much smaller proline increase was shown on the second and third dates. However, on the second date, the elevated proline level was maintained in plants grown with higher NaCl and potassium doses. An increase in salinity resulted in a rise in P5CS activity (Figure 8B). The largest increases dependent on the salt dose were revealed on the first date. The potassium dose did not affect the activity of this enzyme. Salinity affected the decrease in PDH activity (Figure 8C). This decrease was dependent on the date and salt. A greater decrease was found with higher salinity on the third date. This decrease was slightly alleviated by the increased potassium dose in plants grown with lower salinity on the first and second dates.

## 3. Discussion

Crops’ resistance to salinity is an important feature affecting the productivity and final yield. It is not only the ability to survive stress but also the ability to reduce the adverse effects of stress on growth and yielding. The traditional methods of assessing plants’ resistance to salinity are based on estimated growth parameters (fresh or dry weight of aboveground parts) and traits of agronomic interest, such as the yield of harvest organs [46]. However, the aesthetic value is also important for horticultural crops, especially ornamental ones [17,47]. Miyamoto et al. [47] proposed a multipurpose method of evaluating the salt stress resistance of horticultural crops. This method is based on establishing the electrical conductivity (EC) of the soil solution, which results in a 25 to 50% reduction in growth or causes at least 25% damage to leaves. Plants are classified into five categories: sensitive (0–3 dS m^−1^), moderately sensitive (3–6 dS m^−1^), moderately tolerant (6–8 dS m^−1^), tolerant (8–10 dS m^−1^), and highly tolerant (10 dS m^−1^). According to this criterion, *Zinnia elegans* ‘Lilliput’ belongs to the category of sensitive plants, as it has over 25% reduction in most growth parameters at EC 1.1–1.2 dS·m^−1^. To test salt resistance in ornamental plants, de Oliviera [17] proposed another method that considers both growth data and visual analyses. The cumulative reduction in the shoot biomass production and the overall appearance of plants (chlorophyll index, flower production) were used to estimate the ornamental index (Orn index) of four ornamental species. The results confirmed that growth inhibition is inadequate for measuring salinity resistance in all ornamental plants. In some plants, slight or moderate reductions in growth can be compensated by morphological traits that favor the aesthetical quality [18]. For example, *Catharanthus roseus* showed a rise in flower production at the salinity of EC 3.0 dS m^−1^, even with a 10% reduction in shoot dry matter and a 15% reduction in the chlorophyll index [17]. The increase in flower production and about a 20% decrease in the chlorophyll index and shoot dry matter at a salinity higher than 8.0 dS m^−1^ were reported in *Ixora coccinea*. This species was considered moderately tolerant (resistant) to salinity [17]. So, the green color of leaves and flower production (plant appearance) and the assessment of the degree of growth reduction should be recommended to evaluate salinity resistance in ornamental species, especially in garden flowers of commercial interest [17]. According to the criteria proposed by Oliviera et al. [17], our results show that *Zinnia elegans* ‘Lilliput’ appears to be sensitive to salinity. It demonstrated a more than 25% reduction in most growth parameters at EC 1.1–1.2 dS·m^−1^ and a slight decrease in chlorophyll (about 10%) at EC 2.3–2.4 dS·m^−1^. A reduction in the decorative value (flower production, flower diameter) of at least 25% was found at the EC 2.3–2.4 dS·m^−1^. We did not find any adverse effects of salinity on the appearance of leaves, such as the tip and marginal leaf burn in 2016. However, in the preliminary experiment (2015), some initial damage symptoms were observed in plants at EC 2.3 dS·m^−1^, along with typical saline injury such as scorching and necrosis around the leaf margins, leading to the defoliation and death of some plants at EC 3.1 and 3.3 dS·m^−1^.

One of the reasons for the inhibition of leaf growth could be a decrease in their hydration level (RWC), which was noticeable on the first sampling date. Leaf growth results from the irreversible expansion of cells due to water absorption and cell wall extension capacity [48]. The presented studies found an increase in lignins in the cell walls of plants grown using both doses of salt. This increase was the highest on the last date. The reduction in the leaf area increased with the duration of stress. On the other hand, the reduction in RWC on the last date was lower than that on the first date, indicating the involvement of cell wall lignification in limiting leaf growth. Plant cells adjust to salt stress by accumulating lignin and thickening the cell wall [49]. Lignification results in a loss of wall extensibility. As a result, the cell absorbs less water, and growth is inhibited [50]. Cell wall thickening, especially in leaf tissues, has desirable effects on the capacity for turgor maintenance [51]. Moreover, the accumulation of lignin is beneficial for salt-stressed plants because it provides the mechanical strengthening of the cell walls and the protection of membrane integrity [52]. However, salt stress significantly reduced growth parameters in maize without lignin level alteration [53].

The harmful effect of salinity on plants is also caused by the accumulation of toxic sodium ions in cells and disturbances in the uptake of potassium ions [32,54]. This makes it difficult to maintain an optimal potassium (K^+^)–sodium (Na^+^) ratio, which is essential for enzymatic reactions in the cytoplasm, which are necessary for the maintenance of plant growth and yield development [55]. Moreover, an appropriate potassium level is important for maintaining membrane integrity [56]. Our results indicate that such disturbances in the nutritional status of plants (significant increase in leaf sodium and decrease in potassium concentration) caused the reduction in all yield parameters, affected membrane injury, and caused some decline in chlorophyll.

As mentioned in the introduction chapter, the negative effect of salt stress caused by excess sodium chloride concentration may be mitigated by increasing potassium fertilization. In our experiment, the increase in the potassium dose significantly reduced the accumulation of sodium ions in the leaves. There was a slightly smaller but statistically significant increase in potassium levels. A positive effect of the increased potassium dose on membrane integrity was also found. This alleviating effect of the increased potassium dose on cell membrane integrity could be related to the lignification of the walls, leading to their strengthening [52]. A higher dose of potassium had no effect on the chlorophyll content in the leaves of plants grown in the salt-free substrate and at both salinity levels. Similarly, the potassium dose had no impact on the aboveground parts and the number of flowers produced. On the other hand, an increase in potassium decreased the diameter of flowers, the number of lateral shoots, and the leaf area. One reason for leaf growth inhibition by the increased potassium dose could be the increased cell wall lignification. Additionally, in plants growing in the salt-free substrate with an increased dose of potassium, a slight but statistically significant inhibition of leaf growth and an increase in lignin levels were found.

Proline accumulation is a well-known plant strategy for coping with salt stress [57,58]. This amino acid plays an important role in the avoidance of osmotic stress (water deficit) as well as in stress tolerance (water deficit, accumulation of toxic ions) strategies [59,60]. The literature indicates that not only the accumulation of this amino acid but also its metabolism (synthesis, oxidation) may play an important role in stress-coping mechanisms [61]. However, there are limited data on the effect of salt stress on the proline concentration and the mechanism responsible for its accumulation in ornamental plants. Proline accumulation in Jerusalem artichoke plantlets under short-term salinity stress is caused by activating its synthesis and inhibiting its oxidation [62]. We found a significant effect of salinity on the accumulation of free proline in zinnia leaves on the first sampling date at a higher salt dose. Changes in the activity of proline metabolism enzymes indicate that the accumulation of this amino acid resulted from an increase in the activity of the enzyme catabolizing its synthesis (P5CS) and a decrease in the activity of the enzyme responsible for its oxidation (PDH). The proline level did not change at the lower salt dose on the first date. This may have been due to a slight increase in P5CS activity and no change in the activity of PDH involved in proline oxidation. On another date, the proline level did not change significantly under the influence of NaCl doses despite a slight increase in P5CS activity and a decrease in PDH activity. Similarly, in chrysanthemum leaves, proline accumulation decreased with the duration of salinity stress, but it was consistent with the changes in P5CS activity [63]. In cadmium-treated pea plants, proline accumulation corresponded closely with the expression of genes encoding proline synthesis (P5CS) and proline degradation (PDH) enzymes, but the level of accumulated proline was lower in older leaves than in younger ones [64]. In our experiment, leaves were taken from the fourth node (from the top of the plant) on each date. So, on subsequent dates, they were getting older.

Increasing the potassium dose resulted in a significant proline increase in combination with a higher NaCl concentration on the first and second dates and a slightly lower increase on the third date. This greater proline accumulation may also be involved in mitigating damage to cell membranes observed in plants supplemented with higher potassium doses. The involvement of potassium in proline accumulation has already been shown in salt-stressed *Avena sativa*, *Pereskia bleo*, and Pearl Millet [65,66,67]. Potassium-induced proline accumulation in maize was caused by the enhanced conversion of arginine into proline [68]. Given that potassium-induced proline accumulation in salt-stressed zinnia was not triggered by the changes in the activity of the glutamate-proline metabolic pathway (P5CS, PDH), it can be assumed that the synthesis from arginine may have been activated.

In conclusion, our findings indicate that, based on the growth and flowering parameters, *Zinnia elegans* ‘Lilliput’ may be considered sensitive to salinity. Long-term salt stress led to about a 25% reduction in most growth parameters at EC 1.1–1.2 dS·m^−1^ and a similar reduction in the decorative value (amount of flower production, flower diameter) only at EC 2.3–2.4 dS·m^−1^. No signs of damage, such as leaf yellowing and browning, or necrotic spots were observed for any of the used salt concentrations. Salt-induced proline accumulation was the highest at the beginning of stress and was consistent with the change in the activity of P5CS and PDH. Higher potassium doses affected the increase in the K/Na ratio and aided the effective plant adjustment to salinity. Potassium improves *Zinnia elegans’* functions in saline conditions by lessening the accumulation of sodium and chlorine ions, diminishing membrane injury, and triggering proline accumulation. However, potassium did not improve growth and flowering parameters; on the contrary, it affected cell wall lignification in leaves, leading to growth restriction.

## 4. Materials and Methods

### 4.1. Plant Cultivation and Treatment

Three greenhouse independent experiments were conducted during three vegetation seasons (2015, 2016, 2017) from early March to early May. The greenhouse was equipped with a climate computer Hortimax GPK2000; therefore, growth conditions in both experiments were similar. Klasmann high moor peat (pH 3.91) was used to prepare the substrate. Zinnia (’Lilliput’) seeds were sown to the limed peat (pH 6) enriched with multicomponent fertilizer PGMix (0.5g·dm^−3^). Four-week-old seedlings were re-planted into a container filled with 7 dm^3^ of limed peat substrate supplemented with the following final doses of macro- and micronutrients (mg dm^−3^ of the substrate): N-100, P-75, K-150, Ca-1245, Mg-160, Fe-75, Mn-35, Zn-30, Cu-10, B-2, and Mo-1. 

In the first preliminary experiment (2015), plants were cultivated with the following doses of sodium chloride: 0 (control), 0.44, 0.96, 1.47, 1.98, 2.48, and 2.99 g·dm^−3^ of the substrate. The salts were used separately: once at the beginning of the experiment in one complete dose per container. The initial electrical conductivity (EC) of the substrates was 0.3 (control) 0.75, 1.1, 1.67, 2.3, 3.1, and 3.9 dS m^−1^, respectively. The effect of salinity on the growth parameters (height of the main shoot, leaf blade area, fresh weight of the aboveground part of plants, and the diameter of the inflorescence of the main shoot) was evaluated. 

The investigation conducted in 2016 and 2017 aimed to examine the effect of increased potassium doses on the response of zinnia to long-term salinity. Based on the results of a preliminary experiment, we used two doses of salt in these experiments, which had different effects on growth parameters. The experimental scheme for this study was based on a completely randomized design with two levels of sodium chloride (0.96 and 1.98 g·dm^−3^ of the substrate) and two levels of potassium (150 and 300 mg·dm^−3^ of the substrate in form of K_2_SO_4_). The level of other elements was the same as that in the preliminary experiment. Each combination contained five containers filled with the same volume of the substrate and three plants (Figure 9). After adding all the components, the electrical conductivity (EC) of the substrates with lower potassium levels reached 1.1 and 2.3 dS·m^−1^ and 1.2 and 2.4 for lower and higher NaCl doses, respectively. The control combination consisted of plants growing in the substrate with no added salt. The EC of the peat substrate without NaCl addition amounted to 0.3 and 0.4 mS m^−1^ for lower and higher potassium levels, respectively. 

The EC of the growing medium in each experiment was measured using an Orion Benchtop Conductivity Meter (Thermo Electron Corporation). The water content in the substrate was kept at about 60% of the capillary water capacity. The plants were regularly irrigated using water purified by reverse osmosis. There were no holes in the bottom of the containers to prevent water from flowing out.

In the experiment conducted in 2016, the effect of salinity and an increased potassium dose on the content of Na, Cl, and K in leaves and growth parameters was examined. The height of the main shoot, the number of lateral shoots, the fresh weight of the aboveground part of plants, the leaf blade area, the number of flowering inflorescences, and the diameter of the inflorescence of the main shoot as well as the Na, Cl, and K concentrations were measured at the end of the experiment (after 49 days of growth in the saline substrate). In the next experiment (2017), the effect of salinity and an increased potassium dose on biochemical and physiological parameters were evaluated. Plant material was harvested three times during growth in the saline substrate (after 21, 35, and 49 days) in order to estimate the leaf water content, proline content, activity of Δ^1^-pyrroline-5-carboxylate synthetase (P5CS), and activity of proline dehydrogenase (PDH). At the end of this experiment (last harvest), the membrane injury index, chlorophyll content, and lignin level in the cell wall were also estimated.

### 4.2. Water Loss Measurements 

The total amount of water lost in the containers via transpiration and evaporation from the substrate surface was estimated. The containers were weighed every two or three days, and the water losses were supplemented to maintain soil moisture at about 60% of the capillary water capacity. On other days, water losses were supplemented by adding the same water volume to all containers. We measured the differences between the weight of containers after watering to 60% of capillary water capacity and their weight before watering. The volume of water used for plant watering in the meantime was also included. The increase in plant biomass during the experiments was taken into account, and the weight of the containers was considered. The results presented are the means from five replications (containers).

### 4.3. Growth Measurements 

At the stage of full blooming (51 day of growth in the saline substrate), the height of the main shoot, the number of lateral shoots, the fresh matter of the aboveground parts, the average area of the leaf blades, the number of flowering inflorescences, and the diameter of the inflorescence of the main shoot were determined for 15 plants in each combination. The leaf area was determined with a Leaf Area Meter CI-202 (CID BioSciences Inc., Camas, WA, USA). For this purpose, leaves located at the fourth node (from the top of the plant) were collected.

### 4.4. Na, Cl, and K Measurements

Fully mature leaves were pre-dried at a temperature of 105 °C for 48 h and ground in a mixer. The plant material was then mineralized with a mixture of H_2_SO_4_ and H_2_O_2_ (2:1). The Na and K concentrations were determined by flame emission spectrophotometry. The plant material was mineralized at a temperature of 500 °C to establish chlorine levels. Next, the residue was dissolved in hot deionized water, and after sedimentation, the content of Cl was determined in a clear solution by the nephelometric method [69]. All analyses were carried out in six biological replicates. Each replicate was derived from a separate sample of randomly chosen plants. The measurement results are expressed as a percentage of dry matter (% DM).

### 4.5. Physiological and Biochemical Parameter Measurements

Two or three fully mature leaves taken from the third node (counted from the top of the plant) of five randomly chosen plants were used for the estimation of relative water content (RWC), chlorophyll, lignin, and proline content, as well as the activity of Δ^1^-pyrroline-5-carboxylate synthetase (P5CS) and proline dehydrogenase (PDH) and the membrane injury index. The tissue water content and the membrane injury index were estimated immediately after harvest. Plant material for estimating other parameters was frozen in liquid nitrogen and stored at 20 °C until analysis. The analyses were carried out using five independent biological replicates. Each replicate was derived from different samples of plant material.

### 4.6. Water Content in Leaves

The water content in leaves was estimated by measuring the relative water content (RWC) according to the standard method developed by Weatherly [70], with some modifications [71]. It was calculated using the following formula: $$RWC=\frac{\mathrm{fresh}\;\mathrm{matter}-\mathrm{dry}\;\mathrm{matter}}{\mathrm{fresh}\;\mathrm{matter}\;\mathrm{at}\;\mathrm{full}\;\mathrm{turgor}-\mathrm{dry}\;\mathrm{mater}}\cdot 100$$

### 4.7. Proline

The proline content was measured according to Bates et al. [72]. The plant material (200 mg FW) was homogenized with 4 cm^3^ of 5% TCA. The homogenate was centrifuged at 5000× *g* for 15 min. The supernatant was used for proline determination by measuring the quantity of the colored reaction product of proline with ninhydrin acid. Absorbance was read at 520 nm. The amount of proline in the sample was calculated from a standard curve and expressed in milligrams per gram of dry matter (mg g^−1^ DM).

### 4.8. Enzyme Extraction and Assays 

Frozen samples (0.5 g) were ground in a chilled mortar and pestle at 4 °C with 2.5 cm^3^ of extraction buffer (50 mM phosphate buffer, pH 7.2, containing 1 mM phenylmethanesulfonyl fluoride (PMSF), 1mM EDTA-K_2_, and 1% PVP). The homogenate was centrifuged at 14,000× *g* for 15 min at 4 °C. The supernatant was used to determine Δ^1^-pyrroline-5-carboxylate synthetase (P5CS) and proline dehydrogenase (PDH).

The activity of P5CS was determined according to Zhang et al. [73], with some modifications. The reaction mixture in a final volume of 1 cm^3^ contained 50 mM glutamic acid, 10 mM ATP, 20 mM MgCl_2_, 100 mM hydroxamate-HCl, 50 mM Tris-HCL (pH 7.0), and 0.1 cm^3^ enzyme extract. This mixture was incubated at 37 °C for 15 min. Next, the reaction was stopped by adding 2 cm^3^ of the stop buffer (2.5% FeCl_3_ (*w*/*v*) and 6% TCA (*w*/*v*) dissolved in 2.5 M HCl). The precipitated proteins were removed by centrifugation at 15,000× *g* for 30 min, and the supernatant’s absorbance was measured at 535 nm. The amount of *γ*-glutamyl hydroxamate produced was calculated from the molar absorption coefficient (ε_534_ = 0.25 mM^−1^ cm^−1^) of the Fe^3+^-hydroxamate complex. The enzyme activity was expressed in nkat ∙ mg^−1^ protein.

PDH activity was determined using a modified Rahnama and Ebrahimzadeh method [74]. The reaction mixture enclosed 100 mM Na CO_3_-HCl buffer (pH 10.3), 20 mM proline, 10 mM NAD^+,^ and 0.100 cm^3^ of enzyme extract in the final volume of 1 cm^3^. The reduction in NAD^+^ was measured by the absorbance increase at 340 nm through/over 3 min. The amount of reduced NAD ^+^ was calculated from the molar absorption coefficient (ε_340_ = 6.2 mM^−1^ cm^−1^). Enzyme activity was expressed in nkat·mg^−1^ protein.

### 4.9. Protein Concentration

The total protein concentration was determined according to Bradford [75], using bovine serum albumin as a standard.

### 4.10. Chlorophyll Content

The total chlorophyll content was estimated according to the Hiscox and Israelstam method [76]. Leaf samples (100 mg) were cut into pieces, and pigments were extracted at 65 °C using 5 cm^3^ of dimethyl sulfoxide (DMSO). The optical density of the extract was measured at 649 and 665 nm. The total chlorophyll content was calculated following the modified Arnon equations [77] and expressed in milligrams per gram of dry matter (mg·g^−1^ DM).

### 4.11. Lignin Determination

Lignin content was measured by a slightly modified Syros et al. method [78]. Leaf samples were air-dried at 70 °C, and 0.1 g of dry matter was extracted three times with 3 cm^3^ 80% (*v*/*v*) ethanol at 80 °C for 1.5 h. Ethanol was decanted, and the residue was extracted with 3 cm^3^ chloroform at 62 °C. Chloroform was removed, and samples were dried at 50 °C. The dried material was digested in 2.6 cm^3^ of 25% (*v*/*v*) acetyl bromide solution in acetic acid containing 2.7% (*v*/*v*) perchloric acid. After 1h, 100 µl of each sample was added to 580 µl of a solution containing 17.24% (*v*/*v*) 2N sodium hydroxide and 82.76% (*v*/*v*) acetic acid, and 20 µL of 7.5 M hydroxylamine hydrochloride was added to ensure termination of the reaction. Finally, the volume was corrected to 2 cm^3^ with acetic acid. The absorbance of samples was recorded at 280 nm. The lignin content was calculated using a linear calibration curve with commercial lignin alkali (Sigma, St. Louis, MO, USA) and expressed in mg·g dry matter (DM).

### 4.12. Membrane Injury Index

The effect of salinity on cell membrane injury was determined according to Premachendra et al. [79], with modifications [71]. Leaf pieces (five pieces with 1.5 cm diameters for one replication) from control plants and plants treated with different NaCl concentrations were washed quickly three times in 10 cm^3^ of deionized water to remove surface electrolytes. Then, leaf pieces were put into a 50 cm^3^ flask, submerged in 10 cm^3^ of deionized water, and kept at 10 °C for 24 h. After warming to 25 °C and shaking, the electrical conductivity of the effusate was measured. Next, tissues were killed by autoclaving for 15 min and cooled down to 25 °C, and the electrical conductivity of the effusate was measured again. Membrane injury (%) was evaluated according to the formula:$$MI=1-\frac{1-(T1/T2)}{1-(C1/C2)\;}\times 100\%$$
where *C*1 and *C*2 represent the conductivity of the control samples before and after autoclaving, respectively; *T*1 and *T*2 represent the conductivity of the samples treated with NaCl before and after autoclaving, respectively.

### 4.13. Statistical Analysis 

All data were analyzed statistically using STATISTICA 13.3 (StatSoft, Inc., Tulsa, OK, USA). The effect of two factors (salinity and potassium) on the growth parameters (15 replicates), the concentration of Na Cl and K (3 replicates) in leaves, the content of chlorophyll and lignin, as well as the membrane injury index (5 replicates) was determined using a two-way analysis of variance (ANOVA). A three-way ANOVA was used to determine whether the salinity, potassium, and estimation date significantly affected the water loss from containers, the RWC, the proline content, and the activity of P5CS and PDH. Because the results concerning some traits (leaf area, diameter of inflorescence, sodium content in leaves, RWC, P5CS, PDH) did not meet the ANOVA assumptions, they were transformed before performing the analysis using the Box–Cox method [80]. A post hoc Tukey’s simultaneous comparison test was performed if significant differences were found between individual means of the treatment groups in each experiment.

## Figures and Tables

**Figure 1 plants-12-01439-f001:**
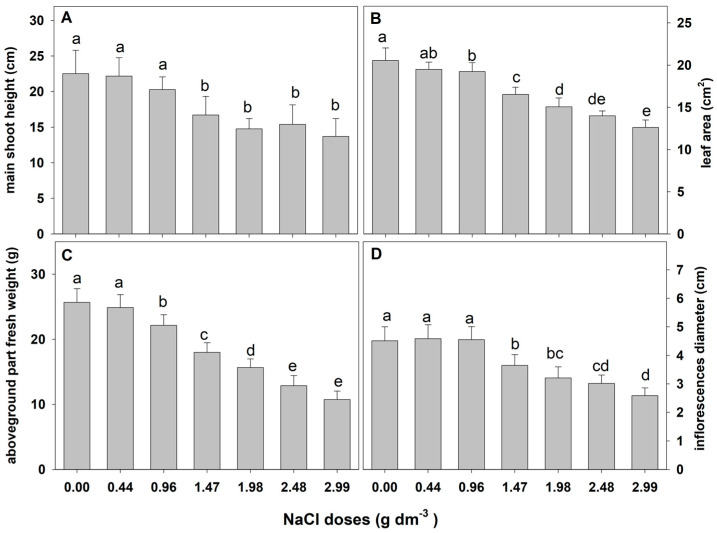
The effect of salinity doses on growth parameters in zinnia. Different letters show statistically significant differences between means. Values are expressed as the means (n = 15) ± standard deviation.

**Figure 2 plants-12-01439-f002:**
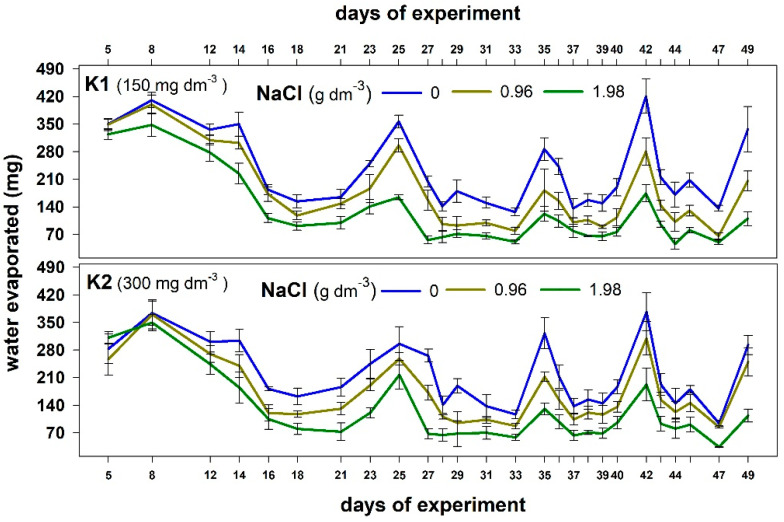
The effect of NaCl doses on water evaporated during the experiment from the containers with plants supplemented with two potassium doses (K1 and K2). Values are expressed as the means (n = 5) ± standard deviation.

**Figure 3 plants-12-01439-f003:**
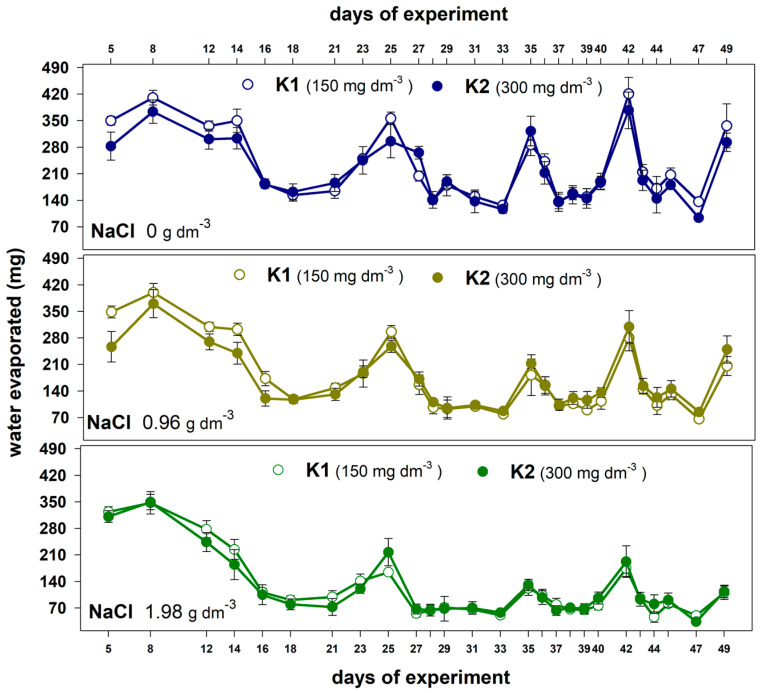
The effect of potassium dose (K1 and K2) on water evaporated during the experiment from the containers with plants grown in the substrate without salt and substrate supplemented with two NaCl doses. Values are expressed as the means (n = 5) ± standard deviation.

**Figure 4 plants-12-01439-f004:**
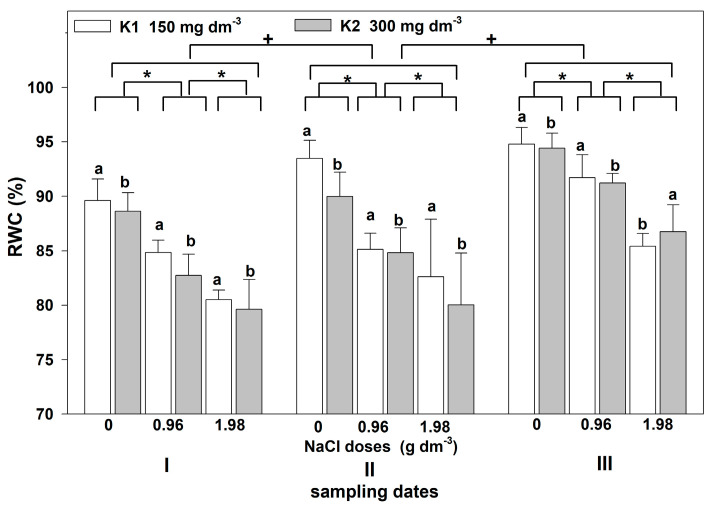
The effect of NaCl doses on RWC in leaves of zinnia grown in the peat substrate supplemented with two potassium doses. Letters a and b show a significant effect of potassium, an asterisk (*) shows a significant effect of salt doses, and + shows a significant effect of the sampling date (I, II, III). Values are expressed as the means (n = 5) ± standard deviation.

**Figure 5 plants-12-01439-f005:**
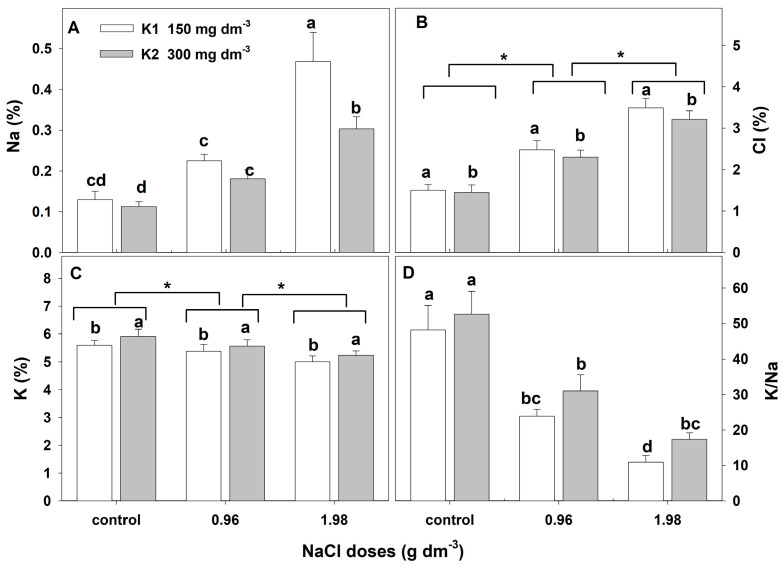
The effect of salinity doses on the concentration (%DM) of Na, Cl, K, and K/Na in the leaves of zinnia grown with two potassium doses. In (**A**,**D**), different letters show statistically significant differences between combinations; in (**B**,**C**), letters a and b show a significant effect of K doses, and an asterisk (*) shows a significant effect of salt doses. Values are expressed as the means (n = 6) ± standard deviation.

**Figure 6 plants-12-01439-f006:**
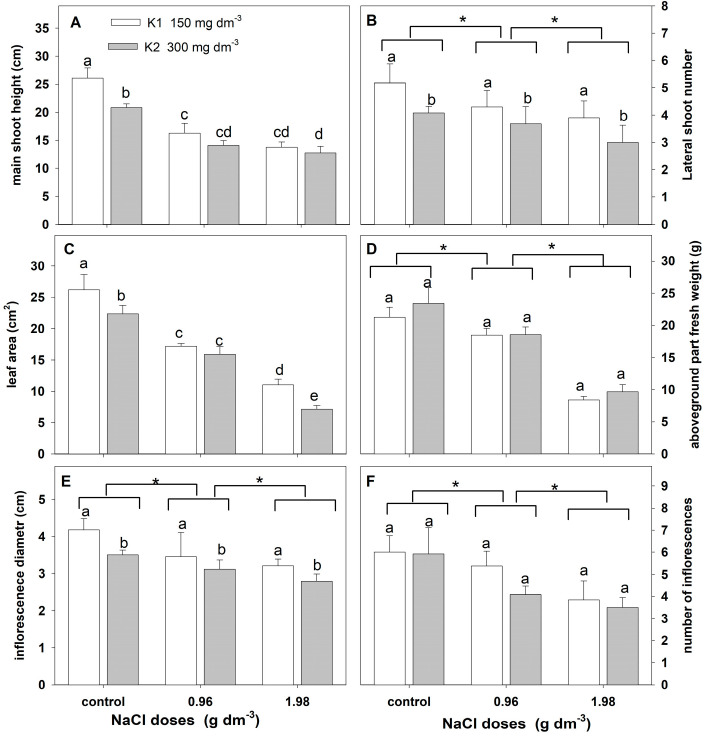
The effect of salinity doses on growth parameters in zinnia grown in the peat substrate supplemented with two potassium doses. In (**A**,**C**), different letters show statistically significant differences between combinations. In (**B**,**D**–**F**), letters a and b show a significant effect of potassium doses, and an asterisk (*) shows a significant effect of salt doses. Values are expressed as the means (n = 15) ± standard deviation.

**Figure 7 plants-12-01439-f007:**
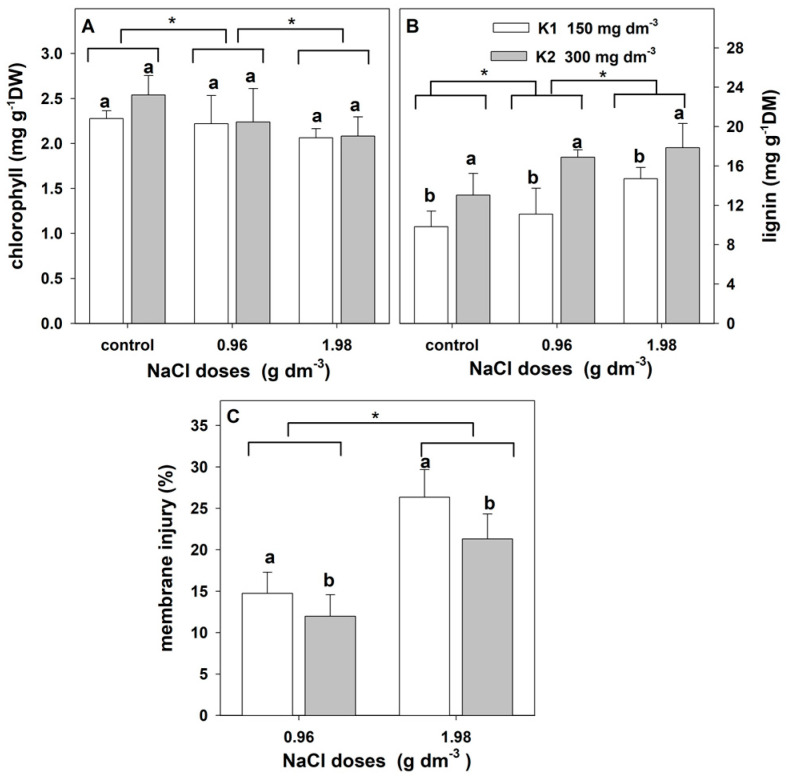
The effect of salinity doses on chlorophyll levels, lignin levels, and membrane injury index in the leaves of zinnia grown in the peat substrate supplemented with two doses of potassium. Letters a and b show a significant effect of K doses, and an asterisk (*) shows a significant effect of salt doses. Values are expressed as the means (n = 5) ± standard deviation.

**Figure 8 plants-12-01439-f008:**
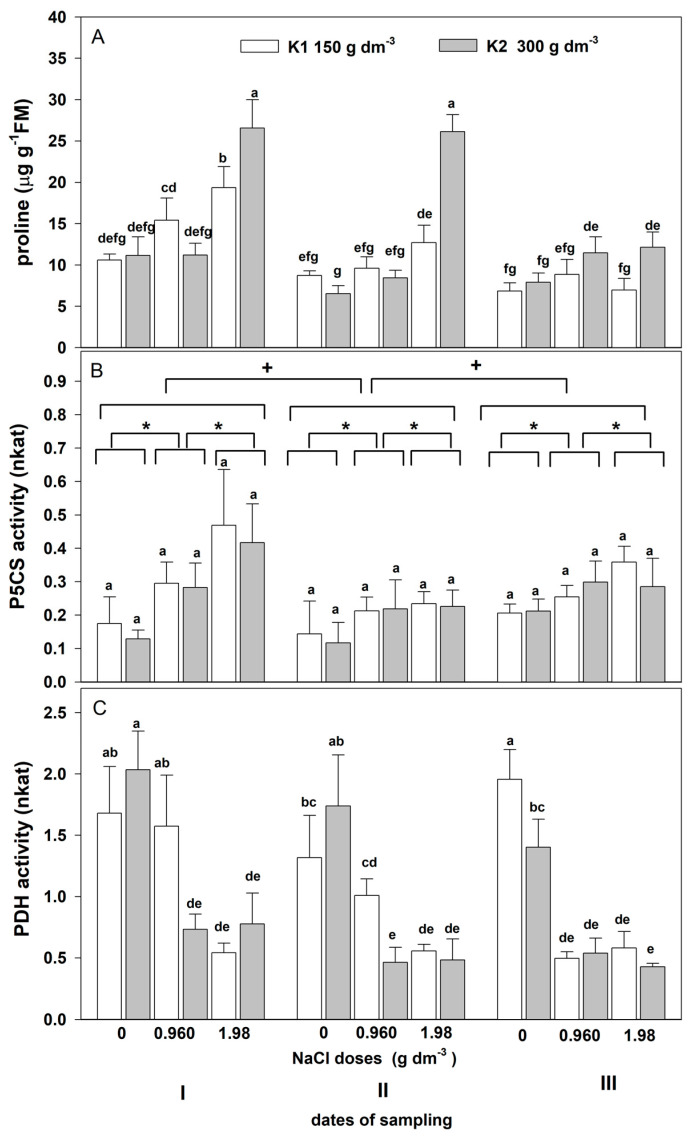
The effect of salinity doses on (**A**) proline content, (**B**) P5CS activity, and (**C**) PDH in the leaves of zinnia grown in the peat substrate with two doses of potassium. In (**A**,**C**), different letters show significant effects of NaCl and K doses and sampling dates. In (**B**), letter “a” shows the lack of a significant effect of the K^+^ level, an asterisk (*) shows a significant effect of salt doses, and **+** shows a significant effect of the sampling dates (I, II, III). Values are expressed as the means (n = 5) ± standard deviation.

**Figure 9 plants-12-01439-f009:**
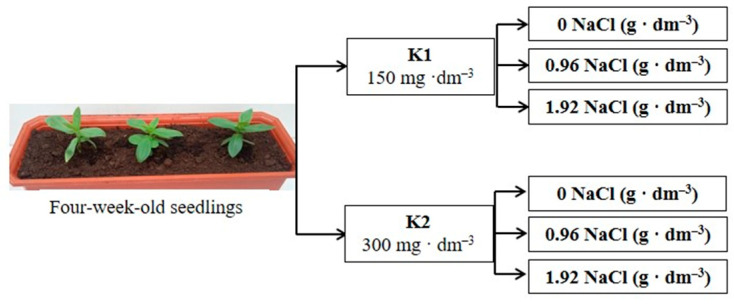
The schematic diagram of experiments; K1 and K2—potassium levels.

**Table 1 plants-12-01439-t001:** One-way ANOVA results for the effect of salt doses on growth parameters.

Treatments	*Df*	Main Shoot Height		Leaf Area		Aboveground Part Fresh Weight		Inflorescence Diameter	
		*F*	*p*	*F*	*p*	*F*	*p*	*F*	*p*
NaCl	6	24.369	**0.0000**	95.89	**0.0000**	130.435	**0.0000**	**41.89**	**0.0000**

**Table 2 plants-12-01439-t002:** ANOVA results for water loss from containers and relative water content (RWC) in zinnia leaves exposed to salinity with two potassium levels in the peat substrate.

Treatments	*Df*	Water Loss		*Df*	RWC	
		*F*	*p*		*F*	*p*
Potassium	1	13.21	**0.0003**	1	5.468	**0.0221**
NaCl	2	1381.32	**0.0000**	2	133.083	**0.0000**
Date	2	399.73	**0.0000**	2	69.872	**0.0000**
Date × potassium	25	5.86	**0.0000**	2	1.889	0.1587
NaCl × potassium	25	8.11	**0.0003**	2	0.925	0.4012
NaCl × date	50	12.97	**0.0000**	4	2.236	0.0735
NaCl × date × potassium	50	2.90	**0.0000**	4	0.977	0.4256

**Table 3 plants-12-01439-t003:** ANOVA results for sodium, chlorine, and potassium concentrations in zinnia leaves exposed to salinity stress with two potassium levels in the peat substrate.

Treatments	*Df*	K		Na		Cl		Na/K	
		*F*	*p*	*F*	*p*	*F*	*p*	*F*	*p*
NaCl	2	25.911	**0.0000**	102.867	**0.0000**	360.356	**0.0000**	**200.123**	**0.0000**
Potassium	1	11.509	**0.0020**	9.628	**0.0002**	8.597	**0.0064**	**16.122**	**0.0004**
NaCl × potassium	2	0.243	0.7856	7.276	**0.0007**	1.477	0.2444	0.288	0.7521

**Table 4 plants-12-01439-t004:** ANOVA results for growth parameters in zinnia exposed to salinity with two potassium doses in the peat substrate.

**Treatments**	** *Df* **	**Main Shoot Height**		**Lateral Shoot Number**		**Leaf Area**	
		*F*	*p*	*F*	*p*	*F*	*p*
NaCl	2	145.342	**0.0000**	7.9817	**0.0022**	385.693	**0.0000**
Potassium	1	29.244	**0.0000**	13.1460	**0.0013**	55.3951	**0.0000**
NaCl × potassium	2	5.9135	**0.0082**	0.3426	0.7133	11.8915	**0.0003**
**Treatments**	** *Df* **	**Aboveground Part Fresh Weight**		**Inflorescence** **Diameter**		**Number of** **Inflorescences**	
		*F*	*p*	*F*	*p*	*F*	*p*
NaCl		180.543	**0.0000**	13.779	**0.0001**	17.996	**0.0000**
Potassium		3.121	0.0900	13.390	**0.0012**	3.378	0.0785
NaCl × potassium		0.4259	0.7224	0.848	0.4408	1.4145	0.2626

**Table 5 plants-12-01439-t005:** ANOVA results for leaf chlorophyll and lignin content and membrane injury index in zinnia exposed to salinity with two potassium levels in the peat substrate.

Treatments	*Df*	Chlorophyll		Lignin		Membrane Injury	
		*F*	*p*	*F*	*p*	*F*	*p*
NaCl	2	4.859	**0.0169**	19.683	**0.0000**	65.818	**0.0000**
Potassium	1	1.323	0.2615	40.929	**0.0000**	9.187	**0.0080**
NaCl × potassium	1	0.871	0.4313	1.877	0.1749	0.766	0.3943

**Table 6 plants-12-01439-t006:** ANOVA results for proline content and P5CS and PDH activity in the leaves of zinnia exposed to salinity with two potassium doses in the peat substrate.

Treatments	*Df*	Proline		P5CS		PDH	
		*F*	*p*	*F*	*p*	*F*	*p*
NaCl	2	129.330	**0.0000**	40.582	**0.0000**	191.364	**0.0000**
Date	2	69.634	**0.0000**	15.330	**0.0000**	17.461	**0.0000**
Potassium	1	32.680	**0.0000**	1.270	0.2635	6.218	**0.0149**
NaCl × date	4	28.069	**0.0000**	4.043	**0.0052**	3.866	**0.0067**
NaCl × potassium	2	41.251	**0.0000**	0.805	0.4512	10.916	**0.0000**
Date × potassium	2	1.689	0.1920	0.233	0.7930	0.974	0.3826
NaCl × date × potassium	4	9.329	**0.0000**	0.441	0.7787	11.756	**0.0000**

## Data Availability

Not applicable.

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
