# Peer review of "Does Potassium Modify the Response of Zinnia (Zinnia elegans Jacq.) to Long-Term Salinity?"

_plants, 2023, doi:10.3390/plants12071439_

Round 1

Reviewer 1 Report

The manuscript submitted by Bandurska et al. deals with the application of potassium and its effect on zinnia plants subjected to saline stress. Although the subject of study is not new, the results obtained by the authors are interesting. The work is very complete, and the results obtained are good. However, the manuscript must be improved before it can be accepted.

In my opinion, the main problem of the study is the concentration of salts used by the authors (1.98 g of NaCl per dm-3 of the substrate, equivalent to 2.4 dS·m-1). This seems to be incongruous, since, in the introduction, the authors mention that a soil is considered saline from 4 dS·m-1 (Lines 37-39). In addition, they also include information regarding the tolerance of zinnia to salinity, with which they confirm that these plants tolerate more than 4 dS·m-1. So, I don't understand why they evaluated EC so low (2.4 dS·m-1)?

General comments:

-Review the correct use of units throughout the manuscript. For example: "ds m-1" should be "dS m-1).

-Include the error bars in the figures.

-Include the number of data used to calculate the mean in the figures.

-Use subscripts and superscripts when necessary. Review the entire manuscript.

Specific comments:

-Delete figure 1 from the introduction. In any case, if you want to use it, maybe you can join the discussion.

-The classification of plant tolerance to salinity seems to be incomplete. Please review and adjust as appropriate. Lines 300-302.

-The authors consider that Zinnia elegans ‘Lilliput’ appears to be moderately resistant to salinity according to Oliviera et al. However, I think it is also important to mention which category it is in according to Miyamoto et al.?. Lines 317-324.

-The authors conclude that Zinnia elegans ‘Lilliput’ may be considered moderately resistant to salinity. However, I believe that this conclusion should be reviewed according to the aforementioned classifications. Line 401.

-Line 419. Describe the characteristics of the substrate.

-Lines 423-424. How many times were NaCl and potassium applied? How much was the total amount applied in the cycle?

-Update used references, especially 17, 34, 40, 41, 49, 68, 69, and 71.

Author Response

Rev. 1

Thank you for your valuable comments and suggestions.

Below are the answers and explanations about the change made. In the manuscript, the changes are marked in red.

In my opinion, the main problem of the study is the concentration of salts used by the authors (1.98 g of NaCl per dm-3 of the substrate, equivalent to 2.4 dS·m-1). This seems to be incongruous, since, in the introduction, the authors mention that soil is considered saline from 4 dS·m-1 (Lines 37-39). In addition, they also include information regarding the tolerance of zinnia to salinity, with which they confirm that these plants tolerate more than 4 dS·m-1. So, I don't understand why they evaluated EC so low (2.4 dS·m-1)?

Literature data show that zinnia (and other ornamental plants) species and cultivars differ in resistance to salinity  In preliminary experiments (data not presented), it was investigated the response of Zinnia elegans ‘Lilliput’ to saline in the range from 0.11 to 3.9 dS m-1. The negative effect of saline on plant height, fresh weight of the above-ground part, leaf area and diameter of the main inflorescence (i.e. the features determining the ornamental value) was found just at the substrate EC of 1.47 dS m-1. This was the basis for selecting the range of salt concentrations used in the experiment.

We have added the relevant information in Introduction.

 General comments:

  • Review the correct use of units throughout the manuscript. For example: "ds m-1" should be "dS m-1). – It has been corrected
  • Include the error bars in the figures. – It has been included
  • Include the number of data used to calculate the mean in the figures. – It has been included
  • Use subscripts and superscripts when necessary. Review the entire manuscript. – It has been chacked

 Specific comments:

  • Delete figure 1 from the introduction. In any case, if you want to use it, maybe you can join the discussion. – It has been deleted
  • The classification of plant tolerance to salinity seems to be incomplete. Please review and adjust as appropriate. Lines 300-302. – It has been supplemented
  • The authors consider that Zinnia elegans ‘Lilliput’ appears to be moderately resistant to salinity according to Oliviera et al. However, I think it is also important to mention which category it is in according to Miyamoto et al.? Lines 317-324. – It has been supplemented.
  • The authors conclude that Zinnia elegans ‘Lilliput’ may be considered moderately resistant to salinity. However, I believe that this conclusion should be reviewed according to the aforementioned classifications. Line 401. – It has been reviewed
  • Line 419. Describe the characteristics of the substrate. – It has been suplemented
  • Lines 423-424. How many times were NaCl and potassium applied? How much was the total amount applied in the cycle? – It has been suplemented in text of manuscript
  • Update used references, especially 17, 34, 40, 41, 49, 68, 69, and 71.
  1. Jardine, M.I. Landscape Architecture Magazine 1924, 14, 270-278. – It was removed
  2. Khalil, M.A.; Amer, F.; Elgabli, M.M. A salinity-fertility interaction study on corn and cotton. Soil Sci. Soc. Am. Proc. 1967, 31, 683-686. – It was removed

  1. Lauter, D.J.; Meiri, A.; Shuali, M. Isoosmotic regulation of cotton and peanut at saline concentration of K and Na. Plant Physiol. 1988, 87, 911-916.
  2. Cerda, A.; Pardines, J.; Botella, M.A.; Martinez, V. Effect of potassium on growth, water relations and the inorganic and organic solute contents for two maize cultivars grown under saline conditions. J. Plant Nutr. 1995, 18, 839-851.

the above pepers (40, 41) have been updated by:

38. Henry, E.E.Y.; Kinsou, E.; Mensah A. C. G.; Komlan, F.A.;Gandonou, C.B. Response of tomato (Lycopersicon esculentum mill.) plants cultivated under salt stress to exogenous application of calcium and potassium. J. Appl. Biosci. 2021, 159, 16363 – 16370 DOI: 10.35759/JABs.159.1

49. 46 Maas, E.V.; Hoffman, G.J. Crop salt tolerance - current assessment. ASCE Journal of Irrigation Drainage Divsion 1977, 10, 115-134. – This paper describes the Maas and Hoffman used to assess resistance based on yield and growth parameters. The was method described some time ago, but it is still valid, especially for agricultural plants. So, this paper is work is always up to date.

71. 70. Weatherly, P.E. Studies in water relation of cotton plants. The measurement of water deficits in leaves. New Phytol. 1950, 49, 81-97. – we used Weatherly’s method to determine RWC, so it is important to cite this paper.

68. Rao, R.C.N.; Krishnasastry, K.S.; Udayakumar, M. Role of potassium in proline metabolism. I. Conversion of precursors into proline under stress conditions in K-sufficient and K-deficient plants. Plant Sci. Lett. 1981, 23, 327-334 DOI:10.1016/0304-4211(81)90044-4. - We did not find any other work on the effect of potassium on proline metabolism, and its conversion from arginine.

  1. Weimberg, R.; Lerner, H.R.; Poljakoff-Mayber, A. A relationship between potassium and proline accumulation in salt-stressed Sorghum bicotor. Physiol. Plant. 1982, 55, 5-10

Updated by:

  1. Ahanger, M. A.; Agarwal, R.; Tomar, N. S.; Shrivastava, M.. Potassium induces positive changes in nitrogen metabolism and antioxidant system of oat (Avena sativa L cultivar Kent). J Plant Inter. 2015, 10, 211-223. DOI:10.1080/17429145.2015.1056260. 
  2. Sulandjari, A.; Sakya T.; Wijayanti R.N. Salinity and potassium fertilizer on growth and proline of the medicinal plant Pereskia bleo. IOP Conf. Ser.: Earth and Environmental Science,vol. 824, 6th International Conference on Climate Change 2021, 25 May, Surakarta, Indonesia. DOI10.1088/1755-1315/824/1/012083
  3. Heidari, M.;   Jamshidi. Effects of Salinity and Potassium Application on Antioxidant Enzyme Activities and Physiological Parameters in Pearl Millet. Agric. Sci. in China 2011, 10, 228-237 DOI: 10.1016/S1671-2927(09)60309-6

Reviewer 2 Report

The paper reports the interaction effect of salt stress and K potassium on Zinnia (an ornamental plant). Salt stress is a major abiotic factor, so the aim is worthy.

The paper is well written but there are some issues that deserves correction.

Introduction.

Overall, the introduction section is too long. For example, the text regrading proline (which include one figure) may be shortened. This text is more appropriate to a review article.

Line 61: “and more” is repeated

MM section

Please include a short indication of the substrate preparation and irrigation details, namely if water was allowed to drain (to avoid salt accumulation). I have some concerns regard the fact that tow different experiments were done. It is not clear that both experiments were running simultaneously ensuring the same growing conditions. This fact has important implications, since the authors performed the statistical analysis considering the data as one, and this not true.

L455-l456: the text is not clear. I suppose that the weight of the containers was considered in the calculations and not corrected.

L460: all plants of the two experiments? Not clear.

L478: how many leaves?

Results

Regarding the effects of salat and K there is some lack of novelty since the mains results are somewhat expected. I suspect that plants did not experience a severe salt adverse effects, explaining why no symptoms were found. Moreover, leaf chlorophyll was not affected (fig 7). Thus, the paper report slight changes to the application of K and salt, which is not a substantial and novel pool of knowledge.

The authors also should consider to rewrite the paper in order to clarify the fact that two experiments were done in two consecutive years, where different growing conditions were possibly observed. My firs recommendation is to reject the paper, but a deep revision  may solve some problems.

Author Response

Rev. 2

Thank you for your valuable comments and suggestions.

Below are the answers and explanations about the change made. In the manuscript, the changes are marked in red.

Introduction.

  • Overall, the introduction section is too long. For example, the text regrading proline (which include one figure) may be shortened. This text is more appropriate to a review article. – It has been shortened
  • Line 61: “and more” is repeated – It has been removed

MM section

  • Please include a short indication of the substrate preparation and irrigation details, namely if water was allowed to drain (to avoid salt accumulation). - There was no way for water to flow out of the container.  There were no holes at the bottom of container. In addition, as it has been described in the paper, watering was performed to 60% of capillary water capacity.
  • I have some concerns regard the fact that two different experiments were done. It is not clear that both experiments were running simultaneously ensuring the same growing conditions. This fact has important implications, since the authors performed the statistical analysis considering the data as one, and this not true. -  In the first experiment were determined  growth parameters  in order to assess resistance to salinity. In the second year, the experiment was repeated and physiological and biochemical parameters were determined. In both experiments, the effect of salinity on growth parameters was clearly visible. In the second experiment, these parameters were not determined because most of the plant material was used to determine physiological and biochemical parameters. However, it was possible to maintain relatively similar conditions because greenhouse was equipped with a Climate computer Hortimax GPK2000.
  • L455-456: the text is not clear. I suppose that the weight of the containers was considered in the calculations and not corrected. – It has been changed
  • L460: all plants of the two experiments? Not clear. - It has been rewritten
  • L478: how many leaves? – It has been supplemented

Results

  • Regarding the effects of salt and K there is some lack of novelty since the mains results are somewhat expected. I suspect that plants did not experience a severe salt adverse effects, explaining why no symptoms were found. Moreover, leaf chlorophyll was not affected (fig 7). Thus, the paper report slight changes to the application of K and salt, which is not a substantial and novel pool of knowledge. - Salinity studies usually focus on high salt doses or high concentrations of NaCl solutions (EC 6-10 dS m-1). We did not want to repeat this pattern because the saline level, typical for urban soils, is usually much lower and falls within the EC range of 2 – 3 dS m-1. No genotype of zinnia with such a high sensitivity to salinity has been described so far.
  • The authors also should consider to rewrite the paper in order to clarify the fact that two experiments were done in two consecutive years, where different growing conditions were possibly observed. My first recommendation is to reject the paper, but a deep revision may solve some problems. The plants were grown under similar conditions as the greenhouse was equipped with a Climate computer Hortimax GPK2000. Growth parameters used to assess resistance to salinity were determined in the first year. In the second year, the experiment was repeated and physiological and biochemical parameters were determined.

Round 2

Reviewer 1 Report

The authors made changes to the manuscript that improved its quality. However, the main problem continues to be the EC that was managed in the study.

The authors mention: "Our preliminary experiments showed that in Zinnia elegans ‘Lilliput’ growth was already 136 restricted at the substrate EC of 1.47 dS m-1." Lines 136-137. Therefore, I suggest that you include the results you mention, and make the necessary adjustments in Materials and Methods, Results and Discussion.

Author Response

Dear Reviewer,

Thank you very much for your suggestion.

We have considered these suggestions and completed the manuscript.

Best Ragard

Hanna Bandurska

Reviewer 2 Report

This new version may be accepted for publucation. No further comments.

Author Response

Dear Reviewer,

Thank you very much for your comments and positive decision.

Best Regard

Hanna Bandurska

Round 3

Reviewer 1 Report

The authors made changes to the manuscript following the suggested comments. Because of this, the manuscript increased in quality.

In my opinion, the manuscript can be accepted for publication in its current form.